# Translational Control of Metabolism and Cell Cycle Progression in Hepatocellular Carcinoma

**DOI:** 10.3390/ijms24054885

**Published:** 2023-03-03

**Authors:** Alessandra Scagliola, Annarita Miluzio, Stefano Biffo

**Affiliations:** 1Istituto Nazionale Genetica Molecolare “Romeo ed Enrica Invernizzi”, 20122 Milano, Italy; 2Department of Biosciences, University of Milan, 20135 Milano, Italy

**Keywords:** eIF4E, eIF6, Non-Alcoholic fatty liver disease (NAFLD), Fatty Acid Synthesis (FAS), Fatty Acid Oxidation (FAO)

## Abstract

The liver is a metabolic hub characterized by high levels of protein synthesis. Eukaryotic initiation factors, eIFs, control the first phase of translation, initiation. Initiation factors are essential for tumor progression and, since they regulate the translation of specific mRNAs downstream of oncogenic signaling cascades, may be druggable. In this review, we address the issue of whether the massive translational machinery of liver cells contributes to liver pathology and to the progression of hepatocellular carcinoma (HCC); it represents a valuable biomarker and druggable target. First, we observe that the common markers of HCC cells, such as phosphorylated ribosomal protein S6, belong to the ribosomal and translational apparatus. This fact is in agreement with observations that demonstrate a huge amplification of the ribosomal machinery during the progression to HCC. Some translation factors, such as eIF4E and eIF6, are then harnessed by oncogenic signaling. In particular, the action of eIF4E and eIF6 is particularly important in HCC when driven by fatty liver pathologies. Indeed, both eIF4E and eIF6 amplify at the translational level the production and accumulation of fatty acids. As it is evident that abnormal levels of these factors drive cancer, we discuss their therapeutic value.

## 1. Introduction

The liver is at the crossroads of two remarkable observations: on one side, it is the mammalian organ with the highest rate of protein synthesis and regulates multiple metabolic processes; on the other, it is the organ with the highest regenerative capability in adults. However, the liver is also the target of hepatocellular carcinoma (HCC), the 6th most common cancer in terms of frequency and the 4th in terms of mortality [1]. What is the relationship between protein synthesis, metabolism, and hepatocellular carcinoma? Clearly, this high capability of protein production is necessary during regeneration, to sustain the cell growth necessary for the G1/S phase transition [2], and it can be harnessed by oncogenic mutations. However, we notice that the mechanistic correlation between protein synthesis and metabolism has not been adequately addressed. Lipid synthesis, nucleotide synthesis and protein synthesis are all anabolic processes and, as such, are addressed in biochemistry or nutritional textbooks. In addition, protein synthesis is characterized by the specifics of the process of translation; consequently, it not only generates mass but also specific information [3]. Thus, the translation of specific mRNAs can result in dramatic biological effects. In this review, we will discuss how translational control exacerbates liver disease, and identify potential therapeutic targets.

## 2. The Liver Is the Body Organ with the Highest Level of Protein Synthesis

The liver is necessary for life and end-stage liver disease is the third most common cause of premature death in Western Europe [4]. Liver function comprises several aspects that make this organ an essential metabolic hub. In short, the normal liver weighs approximately 2.5% of the total body weight but receives 25% of the cardiac output. Accordingly, the liver is the main producer of proteins that are massively secreted in the blood, such as albumin. The liver’s contribution to the blood includes the regulated secretion of hormones such as hepcidin, which is necessary for iron metabolism [5], and thrombopoietin [6], in conjunction with the kidneys, which stimulates platelet generation, as well as others. The liver is connected to the intestine by the common bile duct, where liver-produced digestive enzymes are secreted and brought to the duodenum [7]. The massive production of blood proteins and the digestive activity of the liver imply the synthesis of secreted proteins. Accordingly, liver cells have a massive rough endoplasmic reticulum, used for the synthesis of secreted proteins, which is dynamically assembled according to the body’s needs [8].

One additional feature of liver circulation is the portal system, which directly supplies the blood collected in the intestine to the liver via the portal vein. The portal venous blood contains all the products of digestion that have been absorbed from the gastrointestinal tract. In the liver, a specialized vascular endothelium, which is highly fenestrated, allows the efficient exchange of molecules between the main cellular type of the liver, the hepatocyte, and the portal blood. Absorbed molecules include metabolic intermediates, such as glucose and lipids, and orally absorbed drugs. These molecules are processed in the liver before either being released back into the hepatic veins or stored in the liver for later use. The direct connection to the blood collected in the intestine has implications for the protein apparatus, although in this case, it is less apparent. The intense metabolic activity of the liver, which is associated with preferential exposure to the molecules that are absorbed in the intestine, has posed a great challenge to the evolution of this organ. This exposure to a variety of absorbed molecules and the enormous blood flux also risks the threat of a number of potentially cytotoxic compounds such as alcohol or viruses. As a necessary evolutionary solution, the liver is the only mammalian organ able to massively regenerate following a cytotoxic insult [9]. The regenerative property of the liver has been known since ancient times and is reflected in the myth of Prometheus. Hepatocytes die following chemical or viral damage, but the surviving cells re-enter the cell cycle and can regenerate the entire liver mass in a few days. The amazing regenerative capacity of the liver is evident in rat and mouse models, where two-thirds of the mass can be surgically removed and will grow back in one week [10]. This demand for growth and cell cycle progression is obviously accompanied by the need for a powerful capability to produce proteins.

Feeding increases liver translation by up to 50% [11,12]. We define the general increase of all proteins following stimulation as an increase in global translation and define the mRNA-specific increase as an increase in specific translation. Given that, in normal conditions, the postprandial liver does not regenerate, what are the synthesized proteins? Amino acids induce both global and specific translations [13]. Ribo-seq is one of several technologies allowing analysis at the codon level of translation [14]. In the context of normal liver biology, Ribo-seq has shown that translation elongation rates in the liver are the highest among organs [15]. Translation efficiencies vary across diurnal time and feeding regimen, whereas codon dwell times are highly stable [16]. A study has demonstrated that a subset of genes harboring 5′-terminal oligo pyrimidine (TOP) tracts or translation initiators of short 5′-UTR (TISU) elements encoding proteins involved in translation and mitochondrial activity, respectively, exhibit rhythmic translation that is mainly regulated by feeding [17]. Our lab has shown that specific mRNA translation is stimulated by postprandial insulin [12]. In conclusion, at least three distinct biological activities of the liver require an intense protein synthesis capability: (a) the production of plasma proteins, (b) the regenerative capability, and (c) postprandial biosynthetic activity. All these facts make the liver the organ with the highest rate of protein synthesis of all the body’s organs [18]. The crucial question is whether the high levels of translational machinery in liver cells have an impact on the evolution of diseases, in particular that of liver cancer. We will not discuss the effect of branched amino acids as this has recently been addressed elsewhere [19]. We will focus on the basics of the translational machinery in cancer, and the particular features of liver cancer that are relevant to translational regulation.

## 3. Translation Basics and Its Crosstalk with Cancer

Several reviews of high quality have described in detail the impact of various aspects of translation on cancer development and tumor progression [20,21,22,23]. We will briefly summarize (a) some mechanisms of translational control, (b) nodes where translational mechanisms crosstalk with oncogenic signatures, (c) the impact of ribosome biogenesis, and (d) specific HCC mutations that act on the translational machinery.

### 3.1. The Complex Basics of Translation

In recent years, studies have promoted the concept that translational control is a major, if not the most important, regulator of gene expression [24]. According to the central dogma of molecular biology, translation is the second step of gene expression and consists of the decoding of an mRNA into a protein [25]. For decades, translation has been considered an energy-consuming and totally passive step that faithfully converted each mRNA into a protein. The progressive accumulation of evidence, via the combination of “omics” and individual studies, has demonstrated that the relationship between mRNA and protein levels is, in reality, rather poor [26]. Translation can be divided into four phases, initiation, elongation, termination, and recycling. For a given mRNA, initiation is the rate-limiting phase [21]. The discrepancy between mRNA levels and protein levels is, therefore, due to the action of initiation factors. Initiation is controlled by eukaryotic initiation factors (eIFs): each eIF performs a mechanistic step, under the control of signaling pathways. A crucial concept is that untranslated region sequences (UTRs) on the mRNA regulate the sensitivity to eIF activity. Notably, mRNAs can have very long UTRs and even transcript isoforms, with different UTRs that confer differential translational activity [27,28]. In addition, eIF activity is controlled by signaling pathways [29]. It is, therefore, the interplay between eIFs, mRNA sequences, and signaling pathways that generates the specificity and flexibility of translational control.

In the first step, the ribosomal 40S subunit binds the ternary complex formed by tRNA_i_^Met^, GTP, eIF2, to form 43S (Figure 1). This step is limited by four independent eIF2α kinases activated by several stresses [21]. The formation of 43S can be also stimulated by oncogenic signaling, as exemplified by PI3K-mTOR [30]. Then, 43S binds mRNA to form the 48S complex. The formation of 43S is controlled, mainly, by eIF4F assembly under the PI3K/mTORc1 and ERK-Mnk(s) signaling cascades [22]. Lastly, 48S binds a free 60S subunit to form 80S. The availability of 60S is controlled by eIF6 [31]. Notably, the steps so far described mainly refer to canonical cap-dependent translation that accounts for most of the translation (Figure 1). However, cap-independent [22] or non-canonical cap-dependent mechanisms have recently been discovered [32]. eIF3 is a multiprotein complex that exists in several subcomplexes and participates in several steps of translation [33]. eIF3d acts as a non-canonical 5′ cap-binding protein that is activated in response to metabolic stress in human cells [32]. m(6)A RNA modification in the 5′-UTR stimulates cap-independent translation by the recruitment of the initiation factor, eIF3 [34]. eIF3a and -b facilitate the assembly of the translation-initiation complex and promote the translation of over 4000 mRNA transcripts [35], exploiting m(6)A modification of the mRNA. Other characterized eIFs are eIF1, eIF1A, eIF2B complex, eIF5, and eIF5B [36]. eIF1 and eIF1A have a role in the selection of start codons [37]. Notably, in the context of start codon selection, although the AUG codon in the Kozak context is considered the classical start codon [38], relatively high efficiency can also be given by cognate start codons [39], thus greatly increasing alternative products and regulation. eIF2B acts as a regulator of eIF2 activity [40,41]. eIF5, with the assistance of eIF5B, catalyzes the hydrolysis of GTP bound to the 40S ribosomal initiation complex, with the subsequent joining of a 60S ribosomal subunit resulting in the release of eIF2 and the guanine nucleotide [42]. In short, the specific activity of translation factors accounts for specific gene expression, and the number of molecular mechanisms so far identified is probably only a small part of all the possible ones. As we will see, some of these initiation factors play a dominant role in tumorigenesis and cancer progression, both inside and outside of the liver.

### 3.2. Translation Factors and Cancer, a Connection True for Many Tumor Types

In human cancers, the upregulation of several members of the translational machinery is associated with reduced survival [43]. The simple explanation for all these findings is that several oncogenic mRNAs (e.g., cyclins [44], proangiogenic factors [45], the regulators of metabolism [46], and immune modulators [47]) are regulated at the level of translation [22]. Specific genetic evidence for the essential role of initiation factors in cancer progression has been obtained for some of them. The gene dosage reduction of eIF6 greatly impairs oncogene-induced mortality [48] and the translation of 5′UTR with G/C-rich regions and uORFs [12]. Several elements of the eIF4F complex are essential for malignancy [49]. eIF4F consists of eIF4A1, eIF4E, and eIF4G. eIF4E, as part of the eIF4F complex, promotes the recruitment of the 40S ribosomal subunit by interacting with the 5′ terminus of the mRNA. eIF4E levels are rate-limiting for cancer development, as shown by the fact that in mice, a reduced dosage of eIF4E, while compatible with normal development and global protein synthesis, significantly impeded cellular transformation through its action on specific 5′UTRs [50]. The 40S-eIF4F complex scans the 5′-untranslated region (UTR) for the AUG initiation codon. Notably, ribosomes have a weak capacity to unwind mRNA secondary structures, while eIF4A1 has the ability to unwind stable secondary structures in the 5′-UTR during scanning. Given the fact that several structured 5′UTRs encode for oncogenic mRNAs, eIF4A1 is essential for tumorigenesis [51]. The 4E-BPs are negative regulators of eIF4E (Figure 1) that are inactivated by mTORc1 phosphorylation; the knock-in of 4E-BP phosphomutants reduces the tumor burden [52]. Consistently, pathways that converge on translation are mutated in cancer cells. The Myc oncogene acts as a global activator of the entire ribosomal machinery [53]. The PI3K-mTOR and the RAS-ERK are nutrient-sensing pathways almost invariantly activated in cancer that play prominent roles in translational control [21,22,29]. The complexity with which signaling pathways converge on the translational machinery has been described in detail in an earlier paper [29]. In short, oncogenic mutations must take control of specific translation factors in order to be effective.

### 3.3. The Nucleolus as an Additional Site of Tumorigenesis

eIF6 is a specific translation factor that is also essential for ribosome biogenesis [54]. Ribosomes are assembled in the nucleolus through a complicated series of events that include rRNA synthesis and the nuclear transport of ribosomal proteins, which are then assembled on the rRNA with the assistance of more than 100 trans-acting factors in ribosome biogenesis. Details of this process and its relevance to cancer have been recently reviewed [55,56]. In short, several reports estimate that ribosomes are rate-limiting for cellular growth. The alterations in nucleolar morphology observed in cancer cells directly reflect the greatly increased ribosome production. Increased ribosome production in cancer cells is caused by the dysregulation of the three RNA polymerases (Pol) by molecular mechanisms, involving major oncogenic and tumor suppressive pathways, such as c-Myc [57,58], mTOR [59], p53 [58], pRb [56], and PTEN [60]. Proof-of-concept that targeted therapies that selectively inhibit ribosomal subunit biogenesis are efficient at killing cancer cells has been obtained; these observations are discussed in detail in Ref. [61]. In addition to the “quantitative” hypothesis, the qualitative hypothesis predicts that tissue-specific alterations in the number of ribosomal proteins may lead to the heterogeneity of ribosomes and oncogenic translation [62]. As we will see, alterations in ribosomal proteins are a prominent feature of liver cancer.

### 3.4. HCC Oncogenic Mutations That Impact on Translation Factors

One important issue is the intersection of the translational machinery with the oncogenic mutations found in the liver. Hepatocellular carcinoma is considered heterogenous; hence, the correlation between the translational machinery and the mutational burden can be variable among patients. However, a few facts can be stated without uncertainty. Most liver cancers occur in a situation where there is a chronic disease, characterized by inflammation and local regeneration [1]. Ribosomal proteins, such as rpS6, are mandatory for liver regeneration [63]; thus, high levels of the ribosomal machinery are essential for the normal reaction of the liver to acute insults. Tumors with a high proliferative index are, perhaps not surprisingly, characterized by the presence of phosphorylated rpS6 as a marker, thereby activating the PI3K-mTORc1 pathway (Figure 1). The relationship between mTOR, eIFs, and HCC has recently been described in detail [64]. The role in the translation of the specific phosphorylation of rpS6 by the mTOR cascade is often debated. Years of research have clearly established that rpS6 is the most prominent phosphorylated substrate after mTORC1 stimulation, but the molecular consequences are far from clear [65]. However, in the liver, rPS6 phosphorylation may contribute to the specific translation of long ORFs [66]. Following this line, hepatocellular carcinomas with a high proliferation rate have a mutation of either RPS6KA3 or TSC1/TSC2. RPSKA3 is a member of the p90 family of ribosomal protein S6 kinases, is a MAP kinase-activated protein kinase 1b, and has, as its major substrates, rPS6 and eIF4B, following stimulation of the RAS-ERK pathway [67]. TSC1/2 mutations constitutively activate the mTORC1 pathway, thus leading to the direct phosphorylation of 4EBPs, and, indirectly, to rPS6 phosphorylation through the p70 family of RSKs. In conclusion, it is evident that all proliferative HCCs have mutations that massively control the eIF4F axis and ribosomal phosphorylation.

As a general rule, all mutations in the growth factor cascade converge on the translational machinery. A common mutation in HCC occurs at the level of the Wnt/β−catenin pathway. Here, an interesting mutation-translation crosstalk involving FGF19 is recurrently amplified in HCC and acts upstream of the Wnt/β−catenin pathway. A large part of FGF19-mediated activation occurs translationally. Some WNT pathway components have long and structured 5′ UTRs, with a high frequency of polypurine sequences folding into either stable G-quadruplexes or stable secondary structures. The FGF-mediated increase in the translation of WNT pathway components is driven by the RNA helicase and a component of the eIF4F complex, eIF4A [68].

Last, but not least, the crosstalk with the mutational landscape includes c-MYC. In human HCC, c-Myc is frequently overexpressed, and high levels of c-Myc are associated with a poor prognosis [69]. In this context, it is very well known that Myc acts as a powerful transcriptional stimulator of multiple members of the ribosomal machinery [70]. An in vivo mouse model of liver cancer shows that MYC overexpression synergizes with mutated KRASG12D to induce an aggressive liver tumor, leading to metastasis formation. Genome-wide ribosomal footprinting revealed alterations in the translation of mRNAs, including programmed death-ligand 1 (PD-L1). Further analysis revealed that PD-L1 translation is repressed in KRASG12D tumors by functional, non-canonical upstream open reading frames (uORFs) in its 5′ untranslated region [71].

## 4. The Role of Ribosomal Factors in HCC Progression

### 4.1. The Ribosome

We will start our survey with the essential machinery of the translational apparatus, ribosomes. In general, and as expected, rRNA and ribosome synthesis are greatly induced during both regeneration and HCC onset [72]. Ribosomes are constituted by ribosomal proteins and rRNA, which are assembled in the nucleolus. Nucleolar size can be assessed via argyrophilic nucleolar organizer staining with AgNOR. The number of AgNOR-stained nucleoli is an indicator of the grade of malignancy and a predictor of the prognosis of patients with HCC without portal vein involvement [73].

Specific factors necessary for ribosome biogenesis and ribosomal proteins may play an additional role in HCC development and malignancy. Treacle ribosome biogenesis factor 1 (TCOF1) is a nucleolar factor that regulates ribosomal DNA (rDNA) transcription in the nucleolus and is mutated in Treacher Collins–Franceschetti syndrome (TCS), a congenital disorder affecting craniofacial development. TCOF1 promotes tumorigenesis and the progression of HCC [74].

RACK1 is a structural protein of 40S ribosomal subunits, originally described as a receptor for activated PKC that is necessary for specific translation [75,76] and dendritic arborization [77]. RACK1 promotes chemoresistance in HCC [78] and the self-renewal of cancer stem cells [79]. One important question is whether the upregulation of ribosomal proteins increases cancer malignancy because it simply augments the growth capability of cells, or whether it changes the specificity of translation. This hotly debated issue has received considerable attention; in general, evidence for the existence of subtle variations in ribosome structure that may affect the translation of specific mRNAs has been obtained in several models and is extensively discussed in Ref. [62]. Recently, the 60S ribosomal protein RPL23 has been shown to be a tumor metastasis driver in HCC via its capability of regulating the mRNA stability and translation of MMP9 [80]. An interesting study in HepG2 cells has identified RPL28 as the key gene involved in drug resistance to Sorafenib [81]. In another classical study, rPL36a was found to be overexpressed in HCC and led to enhanced colony formation [82]. Overall, we can conclude that a global upregulation of the ribosomal machinery is a conditio sine qua non for HCC development. In this respect, it is not surprising that the loss of or functional changes in the two major tumor suppressor proteins, pRB and p53, cause an up-regulation of ribosome biogenesis [83].

This being said, we should also consider that ribosomal proteins are highly abundant, and even if their half-life in cells is heavily regulated by their specific association with ribosomes, i.e., several ribosomal proteins are unstable if not bound to ribosomes, the possibility that ribosomal proteins exert ribosome independent functions cannot be discarded. Ribosomal protein rPL11 interacts with and inhibits HDM2 tumor-suppressor function, thus leading to the stabilization and activation of p53 [84]. This observation is the tip of the iceberg of a number of findings indicating that free ribosomal proteins may impair cancer progression [56]. In conclusion, HCC development and progression strongly depend on an increase in ribosomal capability and the generation of ribosomes that increase the translation of oncogenic mRNAs. However, some free ribosomal proteins are part of tumor suppression circuits that may have evolved under the pressure to avoid the excessive synthesis of ribosomes.

### 4.2. The Translation Factors

The role of some translation factors in the progression and malignancy of HCC is, at first sight, puzzling. As described in the previous paragraphs, most initiation factors perform specific mechanistic steps downstream of oncogenic activation. The classic eIF4F complex is constituted by eIF4A, eIF4G, and eIF4E. The general involvement of eIF4F in the progression of cancer is well established and is part of a complex research area that aims at its pharmacological targeting [20,49,85]. Direct analysis in mice during hepatocarcinogenesis confirmed, as expected, the oncogenic activation of the eIF4F complex. AKT and N-Ras proto-oncogenes in mice require the activation of the 4EBP1/eIF4E and p70S6K/RPS6 axes [86]. However, the link in humans between the expression of eIF4F members and HCC development is not equally impressive. In general, a comparison between the levels of eIF4F members between the normal liver and neoplastic HCC tissue does not lead to evident overexpression/overphosphorylation. This may be due to the relatively high levels of initiation factors already present in the normal tissue. It should be noted that several translation factors have been isolated from the normal mammalian liver that constituted a rich source, as described in Refs. [87,88]. In addition, as discussed later, different types of HCC cancer may present interesting variations [89]. Thus, most translation factors are certainly altered in HCC: first, the degree of overexpression is limited; second, their impact on HCC development must be individually addressed by targeted genetic analysis. Database analysis shows that high eIF6 mRNA levels are dramatically associated with HCC lethality [90]. Indeed, genetic and expression studies show that eIF6 is fundamental for the progression of Non-Alcoholic Fatty Liver Disease (NAFLD) to HCC [91] and the progression of HCC itself [92,93]. We conclude that abnormal translation may be an early event of HCC progression and contributes to its malignancy. Whatever the status of translation factors in HCC, reliable evidence shows that translation is greatly altered in HCC. Ribo-seq analysis has contributed to our knowledge of aberrant translation. Physiologically controlled translation is disrupted in obesity [94] and in hepatocellular carcinoma [95]. In human hepatocellular carcinoma, direct analysis of translated mRNAs reveals that the consensus top 100 translationally up-regulated genes show significant enrichment in the biological processes related to extracellular matrix (ECM) organization and collagen catabolism [95]. These data suggest that abnormal translation is an early step in the oncogenic program.

In general, molecular signatures that mark the different stages of liver disease progression can be identified using transcriptomics studies [96]. These studies include microarray and RNA-Seq analyses, which have defined the transcriptional profiles of liver biopsies, ranging from human obesity to NAFLD patients with different stages of severity [97,98,99,100]. Recently, a broad and detailed RNA-Seq study in patient liver tissue from across the full spectrum of NAFLD and its evolution to HCC has been reported [101]. Interestingly, one of the stronger pathways positively regulated during the evolution from NAFLD to HCC is the KRAS signaling pathway, whereas one that is downregulated is the mTOR pathway. This situation is highly similar to the one we observed at the translational level, characterized by an increase in eIF6, downstream of RAS/PKC, and a decrease in phosphorylated rpS6 S240/244, downstream of mTOR [91]. Indeed, at the transcriptional level, we confirmed that selected eIFs decrease in NAFLD patients compared to obese patients (eIF1, eIF4b, eIF3a), which can possibly be explained by a progressive decrease in the hepatic global translational rate during the worsening of hepatic steatosis to NAFLD [91]. In short, during NAFLD’s evolution to HCC, we observed a marked reduction in translational capability with the notable exception of eIF6 levels, which increase in order to sustain lipid metabolism at the translational level [12]. Once the transition from NAFLD to HCC is completed, a general overexpression of eIFs is found in HCC conditions. Many studies have demonstrated an upregulation of eIFs in HCC samples, both at mRNA and protein levels: eIF4E and EIF4G2 [102], eIF4A3 and eIF5B in HCC cell lines [103,104], eIF3S3 [104], and eIF3I [105]. High EIF4G2 expression indicates a poor prognosis [106].

In general, the mechanism by which the expression of single eIFs increases during tumorigenesis has not been fully addressed. This being said, as described in detail before, several oncogenes act on the translational machinery. In particular, the Myc oncogene plays a major role in the transcriptional upregulation of the translational machinery [53].

The major problem of bulk RNA-Seq studies in the contest of liver disease progression is that RNA is derived from mixed cell populations; therefore, its levels are heavily biased toward hepatocytes, which constitute most of the mass. In other words, it is possible that the activity of translation factors considerably changes in specific cell types involved in disease, but the event is missed in the global cell population. In the last decade, single-cell RNA sequencing (scRNA-seq) has been widely used to define cell-type specific molecular profiles, identifying previously unknown cell sub-populations in normal and diseased livers [107]. One seminal study has investigated hepatic injury in the context of human cirrhosis. Using this available single-cell RNA-seq data set, we found that eIF6 mRNA expression levels are higher in two cell lineages derived from human cirrhotic livers, cholangiocytes, and hepatocytes [108]. A similar approach could be used to identify eIF expression in progressive stages of liver disease in different hepatic cell types [109,110], which is certainly an area that deserves further attention. As a note of caution, however, we should remember that mRNA expression does not predict protein levels.

## 5. The Different Involvement of Translation Factors on the Basis of Etiology

### 5.1. Viral-Induced HCC

HCC can have viral and non-viral predisposing factors: alcohol abuse, non-alcoholic fatty liver disease (NAFLD), and viral hepatitis are the main risk factors for HCC development. The viral causes of HCC principally arise from the Hepatitis B virus (HBV) and Hepatitis C virus (HCV). Several notable reviews fully discuss the etiological factors of HCC and the specific features of viral hepatitis-driven HCC [1]. In the context of hepatitis driven by HCC, the host translational machinery is hijacked by the presence in the HCV of a very efficient IRES or internal ribosomal entry site, known as HCV IRES. HCV IRES is a highly structured RNA that mediates cap-independent translation. It is essential for HCV replication, requires eIF3, and has been widely studied since the late 1990s [111,112]. The use of IRES elements circumvents the need for some eukaryotic initiation factors (eIFs) [113]; indeed, the initiation factors eIF2, eIF2A, eIF2D, eIF4A, and eIF4G are not involved in translation that is driven by HCV IRES [114]. In addition to mTOR, eIF3, eIF4, and eIF5 can serve as biomarkers for non- and virus-related HCC [115]. The elevated expression of eIF3H is consistently associated with proliferation, invasion, and tumorigenicity in human hepatocellular carcinoma [116]. A detailed study explored the expression of eIF subunits in 235 cases of virus-related human HCC. Phosphorylated (p)-eIF2α, eIF2α, eIF3B, eIF3D, eIF3J, p-eIF4B, eIF4G, and eIF6 were upregulated in HCV-associated HCC. eIF2α, p-eIF4B, eIF5, and various eIF3 subunits were significantly increased in chronic hepatitis B (HBV)-associated HCC. HCC without a viral background displayed a significant increase for the eIF subunits, p-2α, 3C, 3I, 4E, and 4G [89]. Overall, the data support a model wherein during tumor evolution, the host translational machinery may be inhibited by the stress response and/or progressively adapt to viral infection.

### 5.2. NAFLD-NASH Evolution to HCC

The evidence that patients with HCV-induced cirrhosis continue to have a persistent risk of also developing HCC after HCV eradication underlines the fact that the strongest risk factor for HCC is cirrhosis, regardless of cancer etiology [117]. Unlike viral hepatitis, NAFLD has rapidly become the leading etiology of HCC incidence; its contribution to HCC onset is expected to grow in the next few years owing to the increasing rate of obesity and metabolic syndrome in the West [118]. NAFLD is caused by a build-up of fat in the liver, ranging from the excessive cytoplasmic retention of triglyceride in isolated hepatocytes to steatosis (accumulation of lipid droplets in more than 5% of hepatocytes), without alcohol as a cause. We can summarize NAFLD progression to hepatic failure in four stages: (i) liver fat accumulation; (ii) early NASH (Non-Alcoholic Steatohepatosis), characterized by steatosis, ballooned hepatocytes, and lobular inflammation; (iii) the onset of fibrosis, caused by chronic liver inflammation and injury; (iv) liver cirrhosis, a condition involving a permanently damaged liver in which healthy liver tissue is replaced with scar tissue. Approximately 5–12% of individuals progress over time, from NASH to fibrosis, and thence to hepatic failure, especially when associated with metabolic syndrome or diabetes mellitus. While NAFLD and NASH are considered dynamic diseases able to either reverse or progress, the onset of hepatic fibrosis reflects an irreversible process and is the strongest predictive factor for HCC onset and liver-related mortality [119]. Increasing evidence suggests that NAFLD might be a risk factor for HCC, independently of cirrhosis [120]. Multiple parallel hits that comprise metabolic dysregulations and other hepatic insults have been proposed for the pathogenesis of NAFLD [119]. However, the molecular mechanisms leading to disease progression and liver cancer are not completely clarified.

It is known that in the early steatotic phase of NAFLD, the Fatty Acids (FA) that accumulate in hepatic cells are stored in lipid droplets. However, chronic lipid over-accumulation in the hepatocytes results in an excessive production of FFAs (Free Fatty Acids), which causes cellular metabolic reprogramming and lipotoxicity. The excessive accumulation of these fatty acids increases β-oxidation and ROS production, impairing mitochondrial function and causing oxidative stress [121]. ER stress, autophagy dysregulation, and metabolic and mitochondrial dysfunction cause hepatocyte damage, cell death, and chronic inflammatory hepatic reaction [122]. During hepatic chronic damage, hepatic stellate cells undergo cellular activation, starting to synthesize the extracellular matrix components that promote fibrosis, which are mostly collagen and growth factors. The consequent alteration of the hepatic architecture due to fibrotic niche and hepatic regenerating nodules leads to the establishment of cirrhosis and to permanent liver damage [123]. In addition, premalignant hepatocytes secrete chemokines that interfere with immune surveillance and impair immune-mediated tumor suppression. Thus, besides fibrosis, the impairment of tumor surveillance contributes substantially to cancer onset in HCC, but exactly how an inflammatory microenvironment, altered immune function, and continued liver regeneration contribute to genetic instability and cancer is still poorly understood, rendering specific features of NASH-derived HCC somewhat unclear [124]. In the last few years, original studies have started to clarify the role of translational regulation in immune cells and in tumor-infiltrating immune cells [47,125,126], implying that translation factors could be targeted for novel immunotherapeutic approaches. Importantly, it has been demonstrated that the translational regulation of immune regulators facilitates tumor cell evasion from the immune response to promote HCC progression. In particular, MYC activates PD-L1 translation in response to tumor environment changes, allowing for immune evasion, HCC progression, and metastasis formation [71]. Studies have provided proof-of-concept that a translation inhibitor that reduces eIF4E phosphorylation impairs the aggressiveness of liver cancer in mice, potentially enhancing the anti-tumor immune response [71]. In conclusion, translation may control the local activity of immune cells in both the early and late phases of liver disease.

A remarkable observation is the detection of the crosstalk between translation, lipid metabolism, and HCC progression (Figure 2). Translation and cellular metabolism are closely connected: changes in the translation of specific mRNAs involved in glycolytic, fatty acids and nucleotide synthesis pathways support the cells’ ability to rapidly store energy when there is a burst of growth factors and nutrients and to fuel tumorigenesis [127,128]. Protein synthesis is stimulated by nutrient availability [129]. This biological perspective implies that translation is not a cellular passive mechanism but that translational control of metabolic processes and energy storage could have a role in the onset and evolution of metabolic dysfunction in NAFLD, and, consequently, that specific translation factors could become new therapeutic targets in metabolic disorders. eIF4E dosage is important for the translation of the mRNAs involved in cellular transformation and metabolic fluxes [50]. In response to lipid overload, proteins involved in fat deposition are altered in eIF4E-deficient mice. This is due to the fact that distinct mRNAs involved in lipid metabolic processing and storage are enhanced at the translation level by eIF4E. eIF4E inhibition results in increased fatty acid oxidation, which enhances energy expenditure. The additional inhibition of eIF4E phosphorylation, both genetically and by eFT508, a clinical compound, restrains weight gain following the intake of a high-fat diet [130]. These data favor a mechanism by which hyperactivation of the translational machinery increases lipid-induced damage and the progression to HCC. Importantly, eFT508 treatment is reported to reduce tumor growth in multiple models [131].

eIF6 has a dual function and is necessary for both ribosome biogenesis and translation in the cytoplasm [54]. eIF6 activity is rate-limiting for insulin and growth factor-mediated protein synthesis [12,132]. In mice, eIF6 haploinsufficiency causes less postprandial liver translation, associated with a reduction in blood cholesterol and triglyceride levels, and a deficit in fat deposition in white adipose tissue and liver. Mechanistically, eIF6 activity potentiates the translational reinforcement of de novo lipogenesis, regulating the translational efficiency of mRNAs encoding for lipogenic and adipogenic transcription factors that contain an uORF in their 5′UTR, such as C/EBPβ, C/EBPδ, and ATF4 [12]. This model implies feed-forward anabolic transcriptional reshaping, driven by translation (Figure 2). Analyses of human study databases showed that eIF6 levels increase in NAFLD progression, unlike structural proteins of the small and large ribosomal subunits, while eIF1, eIF4B and eIF3A levels decrease. Genetic eIF6 depletion reduces NAFLD to NASH evolution in mice, impacting obesity, steatosis, and fibrosis progression, and restoring insulin sensitivity. Data-mining analysis showed that eIF6 mRNA levels are dramatically associated with HCC progression and lethality in humans and that eIF6 could be a potential diagnostic and prognostic biomarker for HCC patients [93]. In the context of liver cancer, eIF6 genetic reduction affects the incidence and size of surface HCC nodules in mouse models of NAFLD/NASH rapid progression into HCC and blocks the in vitro growth of HCC spheroids. eIF6 depletion reduces fibrotic areas, proliferating cells and liver tumor markers. Thus, the targeting of eIF6-driven translation hinders NAFLD-HCC progression, interfering with FAS and lipid accumulation and preserving mitochondrial bioenergetic activity and FAO [91]. In conclusion, the increased translation activity of eIF4E and eIF6 generates a specific increase in lipid synthesis and a reduction in lipid oxidation.

Other initiation factors seem essential in the progression of NAFLD. eIF5A acts in multiple phases of the translation process. eIFA is modified by hypusine, a natural amino acid derived from the polyamine spermidine and occurring only in eIF5A [133]. In a recent study, it has been demonstrated that exogenous fatty acids administration decreases hypusination and global eIF5A levels. Reduction of eIF5A hypusination impairs the protein synthesis rate and mitochondrial function. Co-treatment with spermidine, a substrate for eIF5A hypusination, reverts the phenotype. Treatment with spermidine also slows down hepatosteatosis and liver inflammation, damage, and fibrosis in a dietary model of NASH, partially preserving the mitochondrial components. Finally, Zhou and colleagues provided evidence that eIF5A hypusination could be reduced in NASH patients and in mice [134]. The connection between eIF5A and HCC has been well-known for some time, wherein aggressive HCCs are characterized by increased eIF5A activity [135]. The fact that eIF5A activity improves the NASH score before tumor onset and worsens the prognosis after HCC onset is not surprising because it depends on the cellular context, highly proliferative in HCC, versus requiring fatty acid oxidation in preventing NAFLD evolution. In this context, we conclude: (1) that the activity of initiation factors is essential, both for NAFLD evolution to HCC and for HCC progression, marking an evident difference between HCCs driven by viral infection and HCCs driven by lipid accumulation, and (2) eIF6 is the only translation factor consistently upregulated through the transition from NAFLD to HCC, and then, HCC progression.

## 6. Concluding Remarks and Druggability

Since it is evident that translation and the ribosome factor exert a pivotal role in the progression of HCC, can they become therapeutic targets? In general terms, the main factor against the targeting of translation factors is that they are also essential to normal physiological processes. In short, potential limits to the pharmacological targeting of initiation factors include non-specific targeting since many initiation factors are ubiquitously expressed and could potentially affect healthy cells as well. Unintended side effects that could be harmful to the patient can, therefore, arise. Other potential problems are the development of resistance and limited efficacy. This is partly due to the complex nature of mRNA translation and the redundancy of the initiation factors, which makes it difficult to develop drugs that target this process effectively. It should be noted that these limits are common, to a different extent, to multiple strategies. However, we have plenty of evidence that some translational mechanisms are specifically amplified in HCC and play a role in the evolution of the disease from NAFLD to HCC. In general, the targeting of translation factors can either hit the signaling pathways upstream of translation factors or their mechanistic action.

The therapeutic inhibition of the translational machinery is a common and well-known effect of tyrosine kinase inhibitors. The tyrosine kinase inhibitor, sorafenib, was the main systemic drug approved for anti-HCC treatment until the advent of immune checkpoint inhibitors (ICI). Currently, in patients with advanced hepatocellular carcinoma, the combination of an immune checkpoint inhibitor, atezolizumab, with bevacizumab, an antiangiogenic agent, has shown greater benefits and more significant improvements in overall survival and progression-free survival (PFS) than sorafenib [136]. However, the administration of tyrosine kinase inhibitors is still recommended if any contraindications for the treatment exist with first-line therapy. The employed tyrosine kinase inhibitors include sorafenib and lenvatinib, as well as regorafenib and cabozantinib [137]. Sorafenib and its similar compound, regorafenib, are oral multi-kinase inhibitors that target VEGFR2, VEGFR3, PDGFR, c-kit, FLT-3, and RET [138]. Lenvatinib targets the VEGF receptors 1–3, FGF receptors 1–4, PDGF receptor α, RET, and KIT and is an effective inhibitor of tumor angiogenesis [137]. Cabozantinib is a broad-spectrum tyrosine kinase inhibitor [139]. Several studies have addressed the way that tyrosine kinase inhibitors affect translation and ribosome biogenesis. As a general rule, drugs such as sorafenib repress the initiation of translation via inhibition of the mTOR [140] and RAS/ERK pathways [141]. Consistently, the combination of sorafenib with the eIF4E-eIF4G inhibitors 4E1RCat (structural) or 4EGI-1 (competitive) synergistically inhibits the cell viability and colony-formation ability of HCC cells [142]. However, the clinical value of the co-inhibition is limited, due to toxicity and resistance. Similarly to the inhibition of translation, tyrosine kinase inhibitors impact ribosome biogenesis, as thoroughly discussed in Ref. [143]. The important role of therapeutic inhibition of ribosome biology is shown by the fact that, for instance, overexpression of the ribosomal protein L28 induces sorafenib resistance [81]. These observations demonstrate the importance of ribosome biogenesis and translation in HCC progression and, at the same time, define the presence of conspicuous adaptive changes linked to clonal variability in the tumor microenvironment [144].

The targeting of intermediate signaling pathways upstream of initiation factors is practically achieved by a variety of drugs that hit either the growth factor cascade or the eIF2α kinase cascade. The rationale has been thoroughly described [20]. A pivotal role is driven by inhibitors of the mTORc1 cascade, such as rapalogs, which have been widely described in several reviews [20,49,145]. As of today, their clinical effect has been modest. Probably the most promising drug target is phosphorylated eIF4E, its pivotal function in tumor progression having been described before [130]. The crucial aspect of eIF4E phosphorylation is that it completely depends on Mnk1/2 kinases and is dispensable for embryonic growth and adult life [146]. Mnk kinase inhibitors such as eFT508 (Tomivosertib) may, therefore, have tumor-specific effects [147]. eFT508 is currently being tested in several clinical trials. The phosphorylation of eIF2α is a key regulatory target for translation control that is important in regulating translation during normal and stress conditions. The regulation of eIF2α phosphorylation is a promising therapeutic, mainly in the context of the treatment of neurological diseases [148].

The mechanistic targeting of initiation factors can be achieved using a variety of compounds. Omacetaxine, previously known as homoharringtonine, inhibits protein synthesis by blocking the formation of the first peptide bond during polypeptide synthesis. Mechanistic studies have established that omacetaxine inhibits global protein synthesis, with a stronger effect on short–half-life proteins. High-throughput expression screening identified the molecular targets for omacetaxine, including key oncogenes such as PLK1 [149]. As a result, omacetaxine represses growth and increases apoptosis in HCC patient-derived organoids, blocking the formation of crucial oncoproteins such as MYC, β-catenin, cyclin D1, and MET [150]. Silvestrol is isolated from plants of the genus *Aglaia* and is a potent inhibitor of translation initiation. Mechanistically, it interacts with polypurine sequences in the 5′-untranslated region (UTR) of selected mRNAs, thereby clamping the RNA substrate into eIF4A and causing inhibition of the translation initiation complex [151]. Early studies applied silvestrol to HCC models, obtaining specific growth inhibition [152]. Zotatifin (eFT226) is a derived eIF4A inhibitor that blocks tumor growth in receptor tyrosine kinase-driven tumors [153]. Clinical trials evaluating its antiviral and antitumor activities are in progress. Recently, our group has shown that eIF6 haploinsufficiency protects from hepatic steatosis fibrosis and the progression to hepatocellular carcinoma in vivo [91]. We isolated a number of inhibitors of eIF6 binding to 60S ribosomal subunits [154] that are effective in reducing the translation of lipogenic transcription factors [91] and the growth of HCC spheroids in vitro [92]. eIF4E-eIF4G complex inhibition can be achieved using 4E1RCat or 4EGI-1 inhibitors [155]. The combination of 4E1RCat or 4EGI-1 with eIF4E-eIF4G inhibitors synergistically inhibited the cell viability and colony formation ability of HCC cells [142].

In conclusion, we provide an overview of the relevance of translational control in hepatocellular carcinoma onset and progression. Targeting “emerging” hallmarks belonging to the translational machinery, even when in combination with current systemic therapies, can be considered an innovative therapeutic avenue against human HCC.

## Figures and Tables

**Figure 1 ijms-24-04885-f001:**
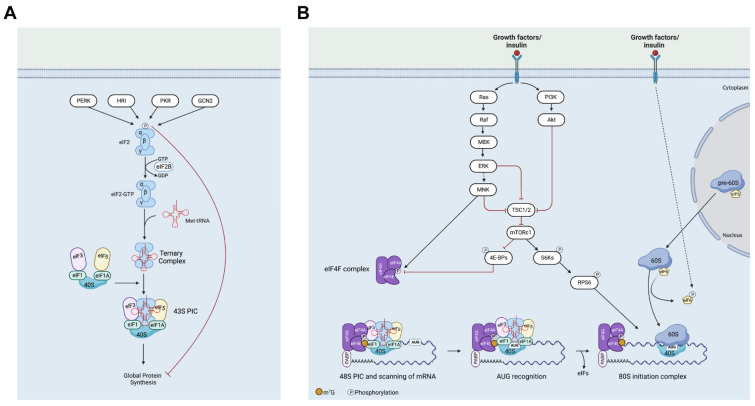
Diagram of the main phases of translational initiation: (**A**) 43S formation. eIF2 ternary complex formation is essential for the initiation of cellular mRNAs. The ternary complex, eIF2, GTP, and Met-tRNA binds the 40S subunit and delivers Met-tRNA to the start codon. Phosphorylation of the α-subunit of eIF2 prevents the formation of the eIF2/GTP/Met-tRNA complex and stops global protein synthesis. Phosphorylation of eIF2α can be achieved by four distinct kinases that can inhibit ternary complex availability by reducing GTP exchange by eIF2B. (**B**) From 48S formation to 80S. The delivery of capped mRNA to the ribosome and scanning to the first start codon is highly dependent on the eIF4F complex, schematically composed of eIF4A helicase, the cap-binding protein eIF4E, and eIF4G. Here, the signaling pathways control initiation at least on two very distinct levels. First, the PI3K-mTORc1 cascade phosphorylates and inactivates a competitor of eIF4E binding to the cap, 4E-BP (multiple isoforms of 4E-BP exist). Second, the RAS-RAF cascade activates Mnk kinase (two isoforms), which phosphorylates eIF4E. Downstream of 48S formation, 60S availability is regulated by eIF6 phosphorylation. The event is complicated by other signaling regulations, such as rPS6 phosphorylation by S6 kinases downstream of mTOR (depicted) or rPS6 kinase downstream of ERK (not depicted). For details, please refer to [29].

**Figure 2 ijms-24-04885-f002:**
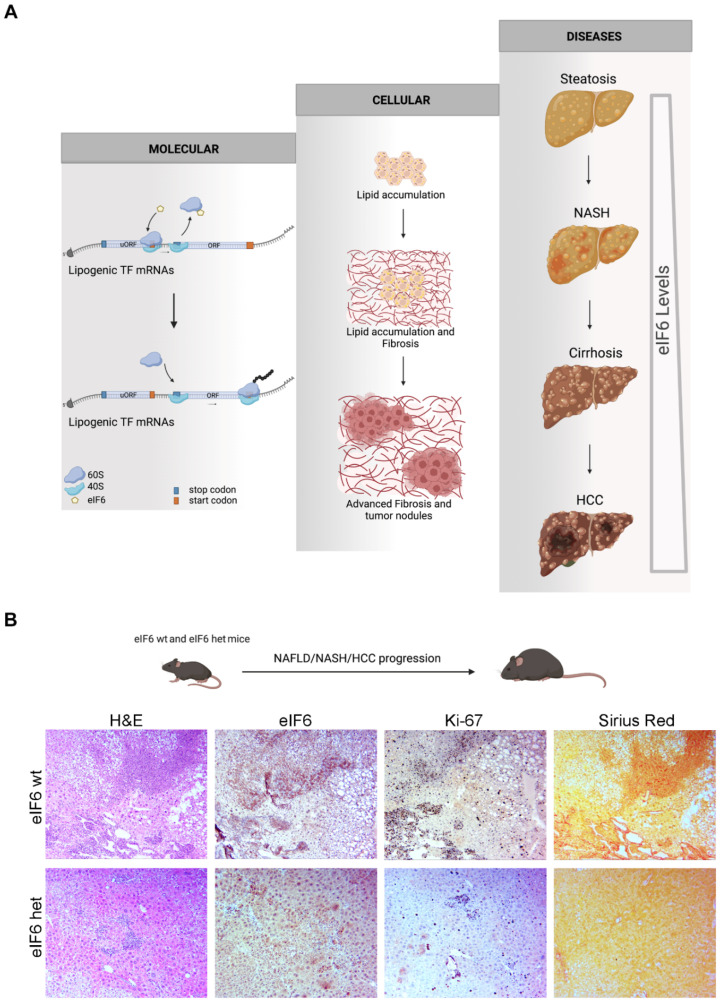
eIF6 levels mark the progression from NAFLD to HCC. (**A**) Increased levels of eIF6 mark this progression, acting as an amplification loop at the molecular and cellular levels of lipid synthesis. Increased eIF6 expression leads to the augmented translation of lipogenic factors [12] containing uORFs (upstream open reading frames) in their 5′UTR. The molecular mechanism leads to more lipid accumulation, premature fibrosis, and tumor nodules. At the expression level, the eIF6 protein augments throughout the process worsening the outcome of HCC [91]. (**B**) During NAFLD/NASH progression to HCC, heterozygous mice for eIF6 have reduced fibrosis, as shown by Sirius Red staining, and less proliferation, as shown by Ki-67 staining [91].

## Data Availability

Not applicable.

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
