# Peer review of "Translational Control of Metabolism and Cell Cycle Progression in Hepatocellular Carcinoma"

_ijms, 2023, doi:10.3390/ijms24054885_

Round 1
Reviewer 1 Report
The manuscript entitled “Translational control of metabolism and cell cycle progression in hepatocellular carcinoma” by Scagliola et al. is a very interesting study focused on the impact of the regulation of translational machinery in the outcome of liver cancer. The interest of the review is dual first describing the impact of ribosomal biogenesis in cancer, as well as the relevance of eIF6 in the induction or progression of NAFLD. The group has published relevant papers in the field. The first line therapy of patients with hepatocellular carcinoma is Atezolizumab and Bevacizumab. However, the administration of tyrosine kinase inhibitors is recommended if exist any contraindications for the treatment with the first line therapy. As several tyrosine kinas inhibitors have shown to impact translation in liver cancer cells, it would be important for readers to incorporate a section describing the different studies addressing the impact of tyrosine kinase inhibitor on protein translation, and the expression on the critical regulators of ribosomal biogenesis.
Author Response
Referee. As several tyrosine kinas inhibitors have shown to impact translation in liver cancer cells, it would be important for readers to incorporate a section describing the different studies addressing the impact of tyrosine kinase inhibitor on protein translation, and the expression on the critical regulators of ribosomal biogenesis
Response: Very relevantcomment. It is evident that tyrosine kinase inhibitors greatly affect the translational machinery and we neglected them. We have added a small paragraph without entering into too many details, given the wealth of data.

Reviewer 2 Report
I was pretty excited to read this mansucript as it directly relates to my research. The review manuscript is well written and cohesive to the topic of interest. The current figures are well thought out and provide the information required.
I was hoping to find a few things in the review which I think are missing and if the information is available in literature, it can help increase the impact of the work.
1. Extensive discusion on RNA-seq based expression of eIFs during disease progression.
2. Extensive discussion on mechanisms of increase in eIFs expression, how it increases with progression of disease.
3. Limitations in targeting eIF for therapy
4. Potential targeting mechanisms
5. Single cell RNA-seq based evidences to identify cell types that have upregulated eIFs in fatty livers/cirrohsis/HCC
Author Response
Ref 1. Extensive discusion on RNA-seq based expression of eIFs during disease progression. 2. Extensive discussion on mechanisms of increase in eIFs expression, how it increases with progression of disease. 5. Single cell RNA-seq based evidences to identify cell types that have upregulated eIFs in fatty livers/cirrohsis/HCC
Answer. These points are indeed important, and define an obvious need for rationalization in this review on the cell type specific/mechanistic aspects of NASH evolution to HCC. On the other hand, especially point 2 is somehow vaste and at the same time unresolved. We wrote a new part to address them (file enclosed)
Ref 3. Limitations in targeting eIF for therapy
Answer. Although they are somehow evident to most readers, we had not identified them. A small paragraph has been added (file enclosed)
Ref 4. Potential targeting mechanisms
The whole "targeting paragraph" has been revised adding the concept of tyrosine kinase inhibitors and re-structuring the logic. Please see the .pdf enclosed for referee 1.
